



# A Comparison of CFAR Object Detection Algorithms for Iceberg Identification in L- and C-band SAR Imagery of the Labrador Sea

Laust Færch[1], Wolfgang Dierking[1,2], Nick Hughes[3], Anthony P. Doulgeris[1]

[1]Center for Integrated Remote Sensing and Forecasting for Arctic Operations, Department of Physics and Technology, UiT—
The Arctic University of Norway, 9019 Tromsø, Norway
[2]Alfred Wegener Institute, Helmholtz Center for Polar and Marine Research, Bussestr. 24, 27570 Bremerhaven, Germany
[3]Norwegian Meteorological Institute, Kirkegårdsvejen, 60, Tromsø NO-9293, Norway

*Correspondence to*: Laust Færch (laust.farch@uit.no)

## Abstract

In this study, we pursue two objectives: first, we compare six different "Constant-False-Alarm-Rate" (CFAR) algorithms for iceberg detection in SAR images, and second, we investigate the effect of radar frequency by comparing the detection performance at C- and L-band. The SAR images were acquired over the Labrador Sea under melting conditions. In an overlapping optical Sentinel-2 image, 492 icebergs were identified in the area. They were used for an assessment of the algorithms' capabilities to accurately detect them in the SAR images and for the determination of the number of false alarms and missed detections. By testing the detectors at varying probability of false alarm (PFA) levels, the optimum PFA for each detector was found. Additionally, we considered the effect of iceberg sizes in relation to image resolution. The results showed that the overall highest accuracy was achieved by applying a Log-normal CFAR detector to the L-band image (F-score of 70.4%), however, only for a narrow range of PFA values. Three of the tested detectors provided high F-scores above 60% over a wider range of PFA values both at L- and C-band. Low F-scores were mainly caused by missed detections of small ($<$ 60 m) and medium-sized (60-120 m) icebergs, with approximately 20-40% of the medium icebergs and 85-90% of small icebergs being missed by all detectors. The iDPolRad detector which is sensitive to volume scattering is less suitable under melting conditions.

## 1 Introduction

Icebergs pose a serious threat to maritime traffic and offshore installations in the Arctic and surrounding regions. As human presence in these areas increases, it becomes more important to develop improved methods for detecting, mapping, tracking, and predicting iceberg occurrences in real-time and over large areas.

Traditionally, iceberg detection in the Northwest Atlantic has been conducted visually by observers on aircrafts of the International Ice Patrol. However, the increased availability of data from satellite Synthetic Aperture Radar (SAR) sensors in the past decades has promoted a move towards automated detection.

SAR is an active instrument that acquires images independent of sunlight and cloud cover conditions. This makes it the preferred sensor for high latitude regions, where cloud cover and a lack of daylight can hinder the use of optical images. In a SAR image, the brightness of each pixel depends on the intensity of the signal that is scattered back from the surface – called the backscattering coefficient. Since different objects exhibit different backscatter characteristics, it is possible to identify targets in the SAR image by looking at the backscatter variations. This identification is further aided by the fact that the SAR sensor can transmit and receive radar pulses at different polarizations, giving rise to additional information about the objects. However, SAR images can be more challenging to analyze compared to optical images for several reasons. Speckle noise, which occurs due to constructive and destructive interference in the radar signal, can make it difficult to identify small features in SAR images. The side-looking geometry of SAR sensors leads to a decrease of the backscattering coefficients with



increasing incidence angle which is most obvious over homogenous targets. A problem that often arises is how to exactly make

a distinction between the backscatter intensity of icebergs and the scattering response of open water or sea ice around them. Despite these issues, SAR images have been widely used for manual iceberg detection. The identification of icebergs in SAR images typically relies on the fact that icebergs tend to have a higher backscatter intensity than open water and certain types of sea ice (Gill, 2001; Sandven et al., 2007; Dierking and Wesche, 2012). In recent years, the development of automated detection schemes has become more and more important. Icebergs can be automatically detected by using either segmentation

(Kim et al., 2011; Tao et al., 2016a; Akbari and Brekke, 2018; Karvonen et al., 2021), or global thresholding approaches (Dierking and Wesche, 2014; Barbat et al., 2019). The most common approach is the application of adaptive thresholding techniques such as the Constant False Alarm Rate (CFAR) detector (Oliver and Quegan, 1997). CFAR detectors are especially valuable for wide-swath SAR images, where large variations in incidence angles make global thresholding techniques difficult to design. Automatic iceberg detection with CFAR has been demonstrated in the past for single-polarization (Power et al.,

2001; Gill, 2001), and for dual- and quad-polarization SAR (Howell et al., 2004; Marino et al., 2016, Zakharov et al., 2017). Regional distributions of icebergs have been mapped using this method, e.g., for Greenland (Buus-Hinkler et al., 2014).

SAR images acquired at C-band (4-8 GHz) are typically employed in operational mapping, e. g. from the European Sentinel-1 mission. Sentinel-1 offers a high revisit interval with daily dual-polarization images over most of the Arctic in its Extra Wide Swath (EW) mode. Through the Copernicus program, Sentinel-1 images are available through a free and open data policy. The

Sentinel-1 mission and its Canadian equivalent, the RADARSAT Constellation Mission (RCM), will continue for at least another decade and will likely be followed by similar missions. C-band SAR is currently being used for iceberg monitoring by, e.g., the International Ice Patrol (IIP) and the Danish Meteorological Institute (DMI).

At the end of this decade, a new L-band (1-2 GHz) SAR mission from ESA, called ROSE-L, will be launched to supplement the C-band Sentinel-1 mission. ROSE-L will also offer regular dual-pol images of the Arctic (Davidson et al., 2021), and it is

anticipated that the L-band data will be a useful complement as the longer wavelength penetrate deeper into snow and ice, revealing the structures underneath (Dierking and Davidson., 2020). Additionally, it is expected that L-band will be less sensitive to sea surface roughness, and therefore will offer a higher contrast between icebergs and sea ice, making detection easier.

Only a few studies on using L-band SAR for iceberg observations have been carried out in the past. Gray and Arsenault (1991)

showed that icebergs cause time-delayed reflections due to internal scattering in airborne L-band SAR images. Marino (2018) tested an iceberg detection algorithm developed for C-band on L-band images with encouraging results. Recently, a study on scattering mechanisms for icebergs in quad-pol L-band SAR images was conducted by Bailey and Marino (2020), and Bailey et al. (2021) later compared various detectors applied on quad-, dual-, and single-pol L-band SAR images. The studies mentioned above give some indication of the dominant scattering mechanisms and detection capabilities of icebergs at L-band.

However, studies comparing C- and L-band for iceberg detection have not been carried out to date and there is still a need to better understand the benefits and limits of both C- and L-band data, and how different detectors perform on the two data types.

A significant challenge in using SAR for iceberg detection is validating the accuracy of detection algorithms. Many studies rely on the information on observed icebergs collected during field campaigns (Willis et al., 1996; Power et al., 2001; Denbina and Collins, 2012), or on visual identification of icebergs in SAR images by experts (Bailey et al., 2021; Marino et al., 2016;

Akbari and Brekke, 2018). Both approaches have limitations. Using field observations of icebergs results in a spatially limited validation dataset, while expert interpretation of SAR images does not account for icebergs that may be present but not visible in the SAR image due to resolution or noise issues. Images from optical remote sensing satellites offer an independent source of validation data but are limited to days with reduced cloud cover. Another requirement is that optical and SAR images must be acquired with a small time-gap between them to avoid icebergs have drifted over long distances between acquisitions.

In this study, we compare six different CFAR detection algorithms and apply them on an overlapping C- and L-band image pair to study the effect of the frequency and tuning of the algorithms on the detection accuracy. To ensure an accurate



comparison, we created a validation dataset using an optical Sentinel-2 image, in which we manually accounted for the iceberg drift occurring between the image acquisitions. The detectors were then assessed not only on their ability to accurately detect the verified icebergs but also on the number of false detections they produced. The novelty of this work is the consistent

comparison between L- and C-band SAR for iceberg detection. Additionally, by using Sentinel-2 data as validation, it was possible to test the detection accuracy as a function of iceberg size. Finally, one of the tested detection algorithms was based on the Wishart likelihood ratio test, which has not been applied before for iceberg detection.

The structure of the paper is as follows. In Section 2, a short introduction to CFAR algorithms for iceberg detection is provided followed by Section 3, which presents the data used, explains the method used for creating validation data, and outlines the

implementation details of the iceberg detection algorithms that we have tested. The paper proceeds with our results in Section 4, and a discussion in Section 5. The paper ends with a conclusion in Section 6.

## 2 Theory

The SAR backscatter intensity from an iceberg mainly arises from surface- and volume-scattering (Power et al, 2001; Bailey and Marino, 2020). These two scattering mechanisms are influenced by several different parameters; some are target-

dependent such as iceberg geometry, temperature, surface roughness, and structure (e.g., the presence of snow, firn or saline layers). Others are sensor-dependent such as incidence angle, frequency, and polarization. Additionally, sensor limitations such as resolution and the presence of speckle noise, further complicate image interpretations, as backscattering returns from icebergs, which mostly cover only a few pixels, might be indistinguishable from intensity variations of speckle. For operational applications, single- or dual-pol are commonly used, but if quad-pol data are available, polarimetric decomposition can be

applied to aid image interpretation (Dierking and Wesche, 2014; Zakharov et al., 2017; Bailey et al., 2021).

It has been observed that icebergs covered by liquid water or wet snow stand out as dark objects against a lighter background of open water. However, in most cases, icebergs exhibit higher backscatter intensities than open water (Power et al., 2001; Wesche and Dierking, 2012). Icebergs are hence typically visible in SAR images as bright spots compared to the relatively darker ocean. Since the backscatter of open water can be highly variable due to its dependence on local wind conditions and

incidence angle, global thresholding techniques are insufficient to detect icebergs. Instead, adaptive methods utilizing the local contrast in backscatter between neighboring pixels are normally employed to distinguish between icebergs and open water.

### 2.1 CFAR Iceberg Detection

A CFAR detector is a type of adaptive thresholding algorithm used to identify objects such as ships or icebergs in SAR images. The algorithm compares the intensity of each pixel under test (PuT) to the local background clutter, and if the pixel value

exceeds a certain threshold, it is marked as an outlier. Clusters of these outliers are assumed to represent objects of interest. The threshold is determined based on the probability density function (PDF) of the local clutter, allowing the CFAR detector to adapt to variations in the background noise (Crisp, 2004).

Accurate CFAR detection thus relies on accurate modeling of the background clutter PDF in SAR images, which is not an easy task. In practice, a handful of models are widely used to estimate sea surface clutter, but their performance depends on

the actual clutter properties, which depend on radar parameters such as frequency. A model that works very well on C-band, might prove inferior on L-band.

The K-distribution is a PDF that has been widely used to model sea surface clutter (Oliver and Quegan, 1997), and CFAR algorithms based on the K-distribution have been used for ship and iceberg detection (Power et al., 2001; Brekke and Anfinsen, 2011; Wesche and Dierking, 2012; Liu, 2018). But due to the complexity of the K-distribution, models based on simpler PDFs

are also commonly used, e.g., the Log-normal distribution (Crisp, 2004; El-Darymli et al., 2013), and Gamma distribution (Gill, 2001; Crisp, 2004; Buus-Hinkler et al., 2014; Tao et al., 2016b).





If the background clutter is accurately modelled, a threshold can be set in such a way, that the probability of falsely triggering the detector – the probability of false alarm (PFA) – is maintained at a constant level. However, in practice, there can be discrepancies between the theoretical PFA, and the actual false alarm rate due to various implementation details; If the window over which the clutter parameters are estimated is too small, it will likely cause the calculated PDF parameters to be biased. If the window is too large, it is more likely to cover nonhomogeneous clutter regions or capture neighboring icebergs which will contaminate the parameter estimation (Tao et al, 2016b). Additionally, CFAR algorithms used for operational detections are often optimized to minimize the computational complexity, which can further degrade the performance. When testing CFAR algorithms for operational monitoring, it is therefore of high importance to inform about the true number of false alarms, e.g., by testing the detector for an area without icebergs.

### 2.1.1 Merging of Multiple Bands

Most models used for estimating the clutter are based on single-channel statistics. For multi-channel data, i.e., the case where several polarizations are available, three distinct detection strategies can be used (Crisp, 2004). (1) The individual channels can be combined into a new single channel, which is then fed to a single-channel detector. This could be achieved by calculating the SPAN (or total power), or by making a new channel consisting of a sum of normalized intensities (Liu, 2015). But channel combinations can also be developed to enhance the contrast between background and target before applying the detector, e.g., by utilizing the polarimetric properties of the target which one wants to detect. One such example is the intensity dual-pol ratio anomaly detector (iDPolRAD) suggested for iceberg detection (Marino et al., 2016). (2) Multi-dimensional detectors based on multivariate PDFs can be applied directly to find outliers based on all channels simultaneously. (3) The far most common approach is simply to apply a single-channel detector to each channel, and then combine the outputs of the resulting detections using Boolean operations.

When merging the output from multiple single-band CFAR detectors into a new channel, it needs to be considered that the PFA of the combined channel will not be the same as the PFA used on the individual bands. If we combine the output from two CFAR filters using a Boolean AND operation, the final product will contain fewer outliers than the number of outliers found by the individual detectors. Similarly, a Boolean OR operation will result in more outliers.

For determination of the combined PFA, the multiplication and addition rules for probabilities can be used.

If we have two detectors, e.g. one applied to the HH polarization, and another to the HV polarization, and assuming that the noise in the HH and HV channel is independent, then the PFA after a Boolean AND operation becomes,

$$PFA(HH \text{ and } HV) = PFA(HH)PFA(HV).$$

If the PFA for the HH and HV channels are equal, we can calculate the PFA needed on the individual channels based on the desired combined PFA as,

$$PFA(HH) = PFA(HV) = \sqrt{PFA(HH \text{ and } HV)}.$$

Similarly, we can calculate the corrected PFA if we are using a Boolean OR operation as,

$$PFA(HH \text{ or } HV) = PFA(HH) + PFA(HV) - PFA(HH)PFA(HV)$$

$$\Rightarrow PFA(HH) = PFA(HV) = 1 \pm \sqrt{1 - PFA(HH \text{ or } HV)},$$

choosing the smallest positive solution, which is also known as the Šidák correction (Abdi, 2007).

This means that if we want a PFA of $10^{-6}$ after combining two detectors using an AND operation, the CFAR detectors applied to the individual channels need to be adapted by applying a PFA of $\sqrt{10^{-6}} = 10^{-3}$. Similarly, if combining using a OR operation, we need an individual PFA of $1 - \sqrt{1 - 10^{-6}} \approx 0.5 \times 10^{-6}$.



**3 Data and Method**

**3.1 Data Description**

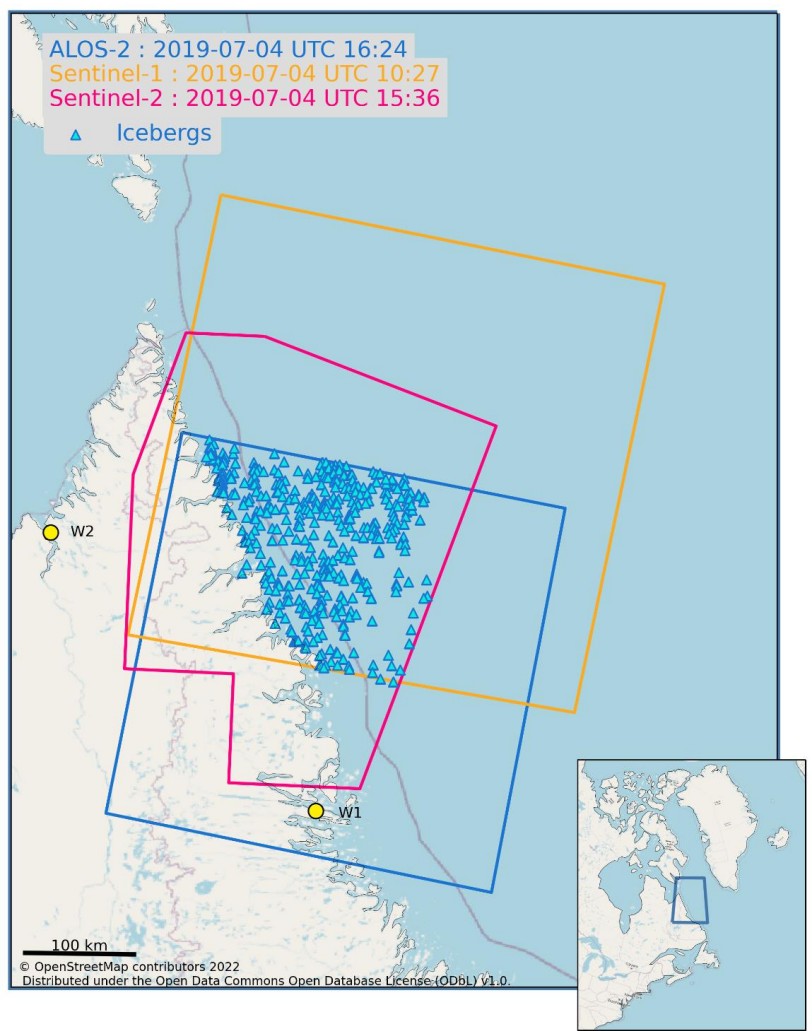

**Figure 1 Overview of the area of interest and data used for the study. The outlines of the optical (red), and SAR (blue and orange) are showed together with the location of the icebergs in the optical image (blue triangles). Meteorological data was downloaded from**
**the weather stations at Nain Airport (W1), and Kangiqsualujjuaq Airport (W2).**

For this study, we selected a test area covering part of the Labrador Sea because of the high density of icebergs in open water and along the coast. This area is also of great interest for operational iceberg charting and is regularly monitored by the International Ice Patrol (Dierking, 2020). An L-band SAR image was acquired by the PALSAR-2 sensor onboard the ALOS-2 satellite. Overlapping Sentinel-1 C-band SAR, and Sentinel-2 optical images were found and downloaded through the

CREOtech Data and Information Access Service (CREODIAS). All three images partially overlapped and were acquired on the same day within a few hours (Figure 1). The ALOS-2 image was delivered in a pre-processed wide beam mode and consisted of dual polarization HH and HV intensity channels. The Sentinel-1 image was acquired in Extra-Wide swath mode (EW) and contained dual-polarization HH and HV intensity channels. The Sentinel-1 image was pre-processed using the



Sentinel Applications Platform (SNAP)[1]. Both SAR images were acquired from similar look geometries, both at a descending

orbit and right looking. The optical Sentinel-2 image was downloaded in level-1C format.

**Table 1 Overview of the data used in the study**

| Sensor | Format | Acquisition Date / Time | Tile(s) / Orbit identifier | Pixel Spacing | Bands | ENL |
|--------|--------|-------------------------|----------------------------|---------------|-------|-----|
| Sentinel-2 | L1C | 2019-07-11 15:26:39 | T20VNJ / R068 T20VNL / R068 T20VPJ / R068 T20VPK / R068 T20VPL / R068 | 10 m | B2, B3, B4, B8 | N/A |
| Sentinel-1 | EW GRDH | 2019-07-11 10:19:33 | - / 028068 | 40 m | HH, HV | 10.7[2] |
| ALOS-2 | WBDR | 2019-07-04 16:24:43 | - / - | 25 m | HH, HV | 15[3] |

Visual inspection of the Sentinel-2 image revealed hundreds of white objects floating in the open water and along the coast.
Although sea ice was spotted in the area two weeks prior to the acquisition of the data used in this study, we expect that most

of the objects in the area are icebergs, since single sea ice floes generally tend to disintegrate faster in open water than icebergs
due to their smaller thickness

An overview of the data is shown in Table 1. Originally, the ALOS-2 image was acquired at a 25-meter pixel spacing, but for
the comparison between C-and L-band both SAR images were resampled to a local polar stereographic coordinate system with
a 40-meter pixel spacing. The resampling was carried out using a Nearest Neighbor interpolation to avoid averaging pixel

intensities. However, the SAR images did still have a different equivalent number of looks (ENL), which should be considered
in the comparison.

Meteorological data from two nearby weather stations were downloaded from the Meteorological Service of Canada (Canada).
At both weather stations the data showed temperatures between 6-15°C, and wind levels between 1-5 m/s during the day the
images were acquired. The weather stations are also shown in Figure 1.

A land-mask was created from Open Street Map land polygons (OpenStreetMap contributors, 2015). A buffer of 500 meters
were added to the mask to avoid any issues with SAR layover or bad geocoding.

### 3.2 CFAR Detectors

Six different CFAR detectors were implemented and tested for this study. Since we are working on dual-pol data, the final
detection needs to include the combination of the HH and HV channels. The selected detectors cover all three detection

strategies outlined in Section 2. Three detectors were based on combining results from single-channels using Boolean logic
(method 3), namely the Log-normal, Gamma, and K detectors Another two detectors were based on transforming the dual-pol
data into a single channel which is better suited for object detection (method 1). These were the normalized intensity sum
(NIS), and the iDPolRAD. Finally, a multidimensional detector (method 2) based on the likelihood ratio test statistic in the
Wishart distribution, was tested as well.

---

[1] The processing steps were: apply orbit file, remove grd-border-noise, thermal noise removal, calibration, and ellipsoid correction to 40 meter pixel spacing.
[2] Sentinel-1 Product Definition (Bourbigot et al., 2016)
[3] ALOS-2 Product Format Description (JAXA, 2012)





### 3.2.1 Log-normal CFAR

The first and most simple detector used was the Log-normal detector (Crisp, 2004; K. El-Darymli et al., 2013). In the Log-normal CFAR detector, it is assumed that the logarithmic transformation from intensity to decibel, normally used for visualizing SAR images, leads to near-Gaussian background clutter. If this is valid, outliers can be detected by employing simple Gaussian statistics, i.e., by comparing the PuT against the average plus some multitude, $k$, of the standard deviation of the background backscatter.

### 3.2.2 Gamma CFAR

The gamma-detector is based on the fact, that under fully developed speckle, the multi-looked background clutter intensity follows a gamma distribution (Oliver and Quegan, 1997; Argenti et al., 2013). Here, the threshold for determining outliers can then be found from the average clutter intensity, and the number of looks, $L$, which is known.

### 3.2.3 K CFAR

The gamma model only accounts for variation due to speckle but in real SAR images, it has been observed that the clutter often exhibits variations in the backscatter in addition to the speckle. These variations, called texture, are attributed to spatial variation of intensity within the area of interest (Oliver and Quegan, 1997; Anfinsen et al., 2009; Doulgeris et al., 2011), and can in some cases falsely trigger a CFAR detector thus leading to a higher false alarm rate. To account for this total speckle variation, clutter models incorporating both speckle and texture have been suggested in the past. The most well-known of these models is the K-distribution. Here, the PDF of the single-band $L$-looked intensity signal, $I$, can be modelled using the mean intensity, $\mu$, the shape parameter (number of looks), $L$, and the order parameter $v$. The disadvantage of this PDF, which is a combination of the gamma function, $\Gamma(z)$, and the Bessel function of the second kind, $K_n(x)$, is that it does not have any closed-form solution. Therefore, a complex numerical integration must be executed to calculate the appropriate threshold.

### 3.2.4 NIS CFAR

The theory behind the normalized intensity sum (NIS) detector is closely related to the principle of the polarimetric whitening filter (PWF) (Novak and Hesse, 1993; Lee and Pottier, 2009). In its original application, the PWF creates a new channel such that the standard deviation to mean ratio is minimized. In the case of dual-polarization intensity data, this new channel can be calculated as a sum of normalized intensities (Liu, 2015) and is therefore referred to as NIS. If we assume that the individual channels, HH, and HV, follow a gamma distribution, the new channel $w$, should also follow a gamma distribution. As such, the CFAR detection for the NIS can be carried out by feeding it into a gamma detector. The method was initially developed for ship detection but has also been tested for iceberg detection (Denbina and Collins, 2014; Bailey et al., 2021).

### 3.2.5 iDPolRAD CFAR

The iDPolRAD was suggested by Marino et al. (2016) specifically for detecting icebergs in sea ice. The detector is based on the observation that icebergs often exhibit a higher cross-polarization and depolarization ratio (cross- over co-polarization) than thin sea ice and open water. This observation is attributed to the fact that radar signals have a larger penetration depth into icebergs than into sea ice and open water, which leads to volume scattering and multiple reflections from within the iceberg volume. This was utilized by designing a detector that is sensitive to pixels with higher cross-polarization and depolarization ratio than their background. Specifically, the algorithm merges the co- and cross-pol channels into a new quantity, that enhances the contrast of pixels with a high volume scattering relative to their background. Icebergs can then be detected in this new quantity, by either employing a global threshold or applying a CFAR detector that is tuned to the PDF of the new quantity.



### 3.2.6 Wishart CFAR

The idea behind the Wishart detector is that under fully developed speckle (Anfinsen et al., 2009; Argenti et al., 2013), the complex amplitude signal of the backscatter follows a circular zero-mean Gaussian distribution, which leads to the complex

polarimetric covariance matrix being Wishart distributed (Goodman, 1963; Conradsen et al., 2003). A test of equality of two complex distributed covariance matrices was suggested by (Conradsen et al., 2003) for change detection applications. A CFAR-like detector for edge detection using this test-statistic was used in (Schou et al., 2003). There, two blocks of equal size, separated by a spatial gap, were used to detect edges in different orientations. This was done by calculating the average covariance matrix for each rectangle and then combining these two covariance matrices into a new channel $Q$, which denotes

the likelihood ratio test statistic. An approximate distribution for $Q$ is known, which can be used to calculate a threshold that corresponds to a specific false alarm rate. The advantages are that this threshold only depends on the block size and the dimensionality of the covariance matrix (number of polarizations). Hence, the threshold only needs to be calculated once per image, which is equivalent to applying a global threshold to the entire $Q$-image. This theory can easily be extended to object detection applications, where a single multi-looked PuT is tested against a larger background. The method is developed for

complex data where knowledge on the full covariance matrix is required. But with minor changes the method also works for intensity data (the block diagonal case in Conradsen et al., 2003; Schou et al., 2003), which will be used for this study.

The main strength of the Wishart detector is that it is multidimensional and can be extended to quad-pol data, without changing the mathematics behind the implementation. Although a contrast enhancement technique based on the test statistic was recently used for highlighting targets with reflection symmetry, suggesting that the method could be used for ship detection (Connetable

et al., 2022), this detector has not been used for iceberg detection until now.

### 3.3 CFAR Implementation Details

All CFAR detectors were implemented using Python 3.8 with the NumPy and Numba libraries (Harris et al., 2020; Lam et al., 2015). Additionally, the SciPy library was used for calculating the statistical parameters needed for the probability density functions (Virtanen et al., 2020). Input and output operations were implemented using the Rasterio library (Gillies et al.).

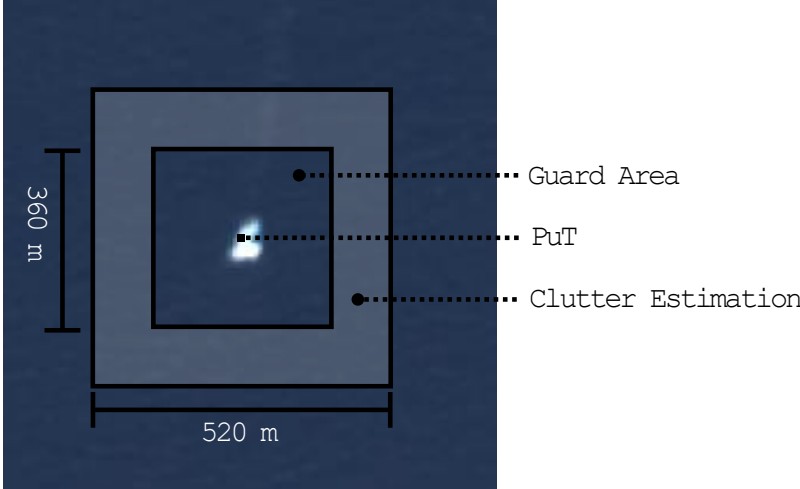


**Figure 2 CFAR sliding window configuration overlayed on the Sentinel-2 image.**

For the Log-normal, Gamma, and K-detector, the detectors were applied to the HH and HV channels individually. The final outliers are then found by combining the outliers from the two channels using a Boolean AND operation. The AND operation was selected to minimize false detections due to single-channel noise, e.g., in the form of speckle. Except for the iDPolRAD

detector, the other detectors were implemented using the window design shown in Figure 2. Here, a guard area of 360 meters



ensures that icebergs shorter than 180 meters will always be excluded from the clutter estimation window regardless of orientation. The window sizes were based on the inspection of the Sentinel-2 image, in which 97% of the icebergs are shorter than 180 meters. Icebergs that are longer than 180 meters but shorter than 360 meters will only partly contaminate the clutter estimation, e.g., when the center of the window is located at the iceberg edges. For sizes ≥ 360 m we found only one iceberg

in the Sentinel-2 image. The size of the outer window of 520 meters, was selected as a trade-off between having a high number of samples for parameter estimation, while avoiding capturing neighboring icebergs in the background estimation.

The iDPolRAD detector was implemented using a test and a training window of 3x3, and 57x57 pixels respectively. These window sizes were chosen based on the suggestions by Marino et al. (2016) and Soldal et al. (2020). In the original paper (Marino et al., 2016), icebergs are detected using a Gaussian-based CFAR detector with an empirically set threshold, since no

analytical expression of the PDF exists for the iDPolRAD transformation. However, this approach is unsatisfactory for the comparison of different detectors in this study. Instead, we have opted on using a method similar to Soldal et al., (2020). Here, a generalized Gamma function proved to be a good fit for the distribution. But since it is computationally very expensive to estimate the parameters for this distribution locally, we decided to fit the generalized Gamma distribution to the iDPolRAD image globally. To avoid skewing the distribution, land was masked, and pixels where the iDPolRAD was smaller than 0, and

larger than 50 times the mean, were excluded for the parameter estimation. Our approach enables us to test the performance of the iDPolRAD detector at varying global thresholds avoiding too long computation times.

Also, the K-detector is computationally expensive when estimating the threshold locally. This is normally solved by using approximations of the original PDF (Oliver and Quegan, 1997; Tunaley, 2010), or by estimating the order parameters regionally on larger image tiles (Liu, 2018). To shorten the computation time for the K-detector, we used pre-computed look-

up tables for the threshold (Brekke, 2009). Threshold values corresponding to the desired PFA-level and the ENL of the image were calculated for 40 different values of the order parameter $v$ on a linear interval between 1 and 20, corresponding to the observed range of $v$ for our data. For values of $v$ larger than 20, the threshold does not change significantly, since large $v$ values correspond with low texture. The order parameter $v$, could be calculated locally for each pixel using the clutter estimation window above, using the method of moments (MoM) as suggested by (Wesche and Dierking, 2012). Based on this order

parameter a suitable threshold was selected from the look-up table for each PuT.

The NIS transformation was calculated by using the window in Figure 2. Here the PuT intensities of the two channels were normalized using the average of the clutter window and then added yielding the normalized intensity sum. A Gamma CFAR detector was then applied to detect outliers in the NIS channel. The Gamma CFAR requires an estimate of the ENL, which has changed after the transformation. The new ENL was estimated from the mean-squared-over-variance ratio. To avoid skewing

the estimation of the ENL due to the presence of outliers, pixels with a NIS above two times the median NIS were excluded.

The Wishart detector was implemented according to (Schou et al., 2003) using the window configuration shown in Figure 2. Here, the covariance matrix of the PuT was compared with the average covariance matrix of the background clutter. Since the Wishart detector is based on a two-sided test statistic, the CFAR filter will highlight both bright and dark features. However, for this study we are only interested in bright outliers, since we only found icebergs with bright radar returns, so outliers that

are darker than the mean of the clutter window were removed.

Each of the 6 detectors was applied to the images 21 times, corresponding to 21 different PFA-levels varying from 10e-21 to 10e-1 on both the Sentinel-1 and ALOS-2 image. Due to issues with numerical stability of the detectors, it was not possible to test the filters at PFA-levels smaller than 1e-21. The land-mask mentioned earlier was applied to the SAR images before CFAR detection to avoid false detections due to land. To limit the noise in the detections, identified objects covering only a single

pixel were removed from the results. Similarly, objects covering more than 500 pixels were also removed from the dataset, since no objects near that size were observed in the Sentinel-2 image, and hence it was assumed that outliers of that size were likely caused by errors in the processing.



The code for the CFAR detectors has been made available on GitHub to allow testing on other SAR images by fellow researchers.

**3.4 Validation Data**

In the Sentinel-2 image icebergs were identified by applying an N-Sigma CFAR detector to the sum of the high-resolution bands; B2, B3, B4, and B8. Pixels brighter than the mean background plus four times the standard deviations were marked as possible icebergs. The results were then manually checked to remove artifacts from clouds and land, and a few icebergs missed by the automatic detection were added. A total of 492 icebergs were detected in the area of interest (AOI) in the optical image. The sizes of the icebergs were then extracted from the Sentinel-2 image and the icebergs were classified according to length (Table 2) using the WMO nomenclature (Dierking, 2020), with the length being the major axis of the icebergs. The average iceberg length was 84 meters, and 97% of the icebergs were shorter than 180 meters.

**Table 2 Classification of icebergs in the area of interest**

| Iceberg Type | Number of Icebergs |
| --- | --- |
| Small (< 60 meters) | 181 |
| Medium (60 – 120 meters) | 175 |
| Large (> 120 meters) | 136 |

Since the optical and the SAR images were acquired at different times, the locations of the icebergs change between the images because ocean currents and wind cause the icebergs to drift between acquisitions. To correct for this drift, icebergs observed in the Sentinel-2 image were manually matched to objects in the SAR images. This matching was carried out in the geographic information systems (GIS) application QGIS 3.10.13, and the process was aided by the fact that on a large scale, icebergs arranged in clusters often drift in similar directions and over a similar distance. Hence, looking at the overall patterns of iceberg clusters across the different images helps determine the drift of individual icebergs. Using this approach, it was possible to create verified drift paths for 336 of the icebergs in the ALOS-2 image, and 270 of the icebergs in the Sentinel-1 image. The reason why fewer icebergs could be manually matched between the Sentinel-1 and Sentinel-2 image, was the larger time difference of approximately 5 hours between their acquisitions, compared to only around one hour difference between the ALOS-2 and Sentinel-2 image. Additionally, a higher noise level of the Sentinel-1 image made matching more difficult, especially for smaller icebergs. The average drift distance was 489 meters between the ALOS-2 and Sentinel-2 acquisitions, and 3953 meters between the Sentinel-2 and Sentinel-1 acquisitions. Most of the icebergs that could not be matched with high confidence between the SAR images and the optical image, were very small, and hence very difficult to visually identify in the SAR images because of their lower resolution and the presence of speckle noise. A linear interpolation method was used to predict the expected drift paths of these icebergs (Figure 3). This linear interpolation method was tested on a subset of the dataset (10%) and showed to give an average distance error of 335 meters for ALOS-2, and 1789 meters for Sentinel-1

A few bright objects in the SAR images that were covered by clouds in the optical data were masked out from the SAR images to avoid counting these as false detections. Similarly, icebergs drifting into the AOI from outside was removed manually from the analysis. Additionally, a single bright object visible in both the ALOS-2 and Sentinel-1 image was interpreted as a ship and removed from the analysis. The object resembling a ship was also recognized in the Sentinel-2 image, but independent AIS confirming this observation could not be found.



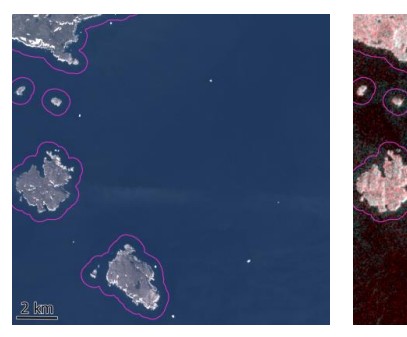
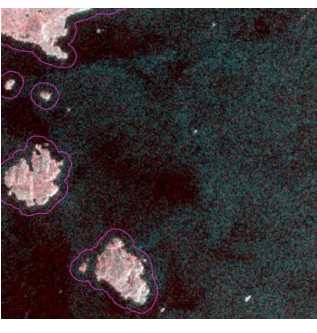
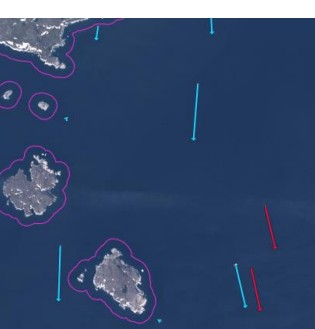

(a) Sentinel-2 Image   (b) C-band image   (c) Drift Vectors

**Figure 3 Subset of the AOI showing the matching between the optical and SAR data. Purple outline indicates areas removed by the land-mask. Several icebergs are visible in the Sentinel-2 RGB image (a). These icebergs have drifted since the Sentinel-1 image was acquired (b). Verified drift paths (light blue) were manually created for icebergs that could be confidently identified in the SAR imagery. Expected drift paths (dark red) were then created using linear interpolation the remaining icebergs. (c).**

Both the verified and expected drift paths were used to validate automatic detections in the SAR images. Objects detected by the CFAR algorithms within a search radius of the expected or verified locations of icebergs were marked as true positives (TP). This search radius was set to 250 meters for the verified drift paths. For the interpolated expected drift paths, the search radius was set equal to the average distance error in the interpolation method; 335 meters in the ALOS-2 image, and 1789 meters in the Sentinel-1 image. If several objects were detected within the search radius, only the nearest object is counted as a true positive, and the rest is interpreted as false positives. If no objects are detected in the search area, it was marked as a false negative (FN). Objects that were not within the search radius of any icebergs were marked as false positives (FP).

## 3.5 Post-Processing

Three different performance measures were used to check the performance of the different detectors. These were Recall, Precision, and F-score defined as,

$$Recall = \frac{TP}{TP + FN}$$

$$Precision = \frac{TP}{TP + FP}$$

$$Fscore = 2\frac{precision \cdot recall}{precision + recall}$$

As such, recall accounts for the probability of detecting the validated icebergs, i.e., how many of the icebergs have been detected. Precision is used to assess the probability of a detected object being an iceberg. As the PFA-level is increasing, each detector is more likely to detect the verified icebergs, but also more likely to make false detections. The overall performance of the detectors is thus a trade-off between these two scores. For marine safety, e.g., a missed detection is more critical than a false detection. For evaluating the overall performance of the detectors, we decided to evaluate missed- and false detections equally by using the F-score.



370    **4 Results**

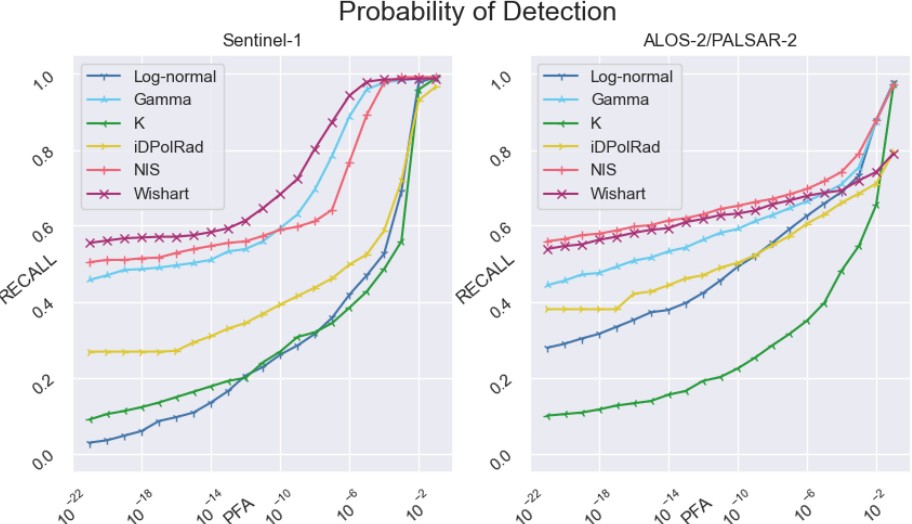

**Figure 4 Recall – the probability of detecting the icebergs.**

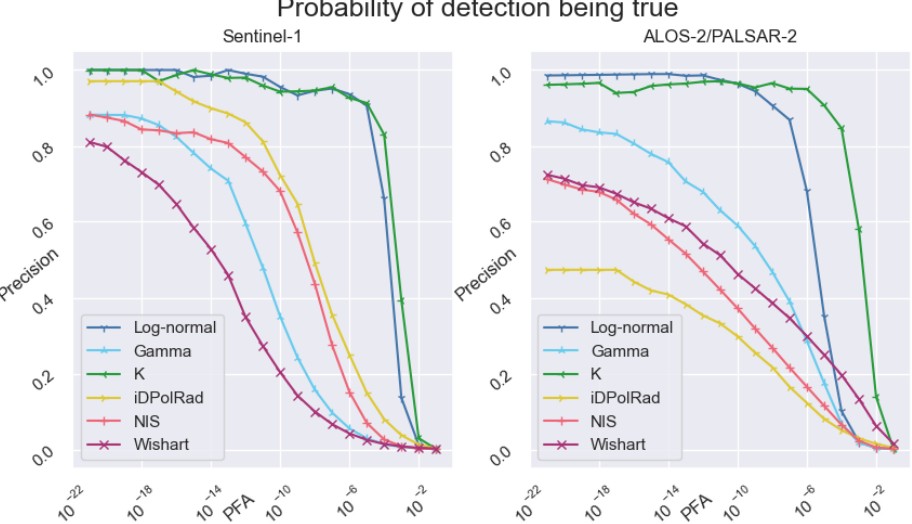

375    **Figure 5 Precision – the probability of a detection being an iceberg**

Recall and precision as a function of increasing PFA-levels are plotted in Figure 4, and Figure 5. For most of the detectors, the shape of the recall function can be divided into two phases. For small and intermediate PFA-levels, the recall is either constant or increasing steadily with increasing PFA. For larger PFA-levels, the recall increases more rapidly. The first phase can be explained by the detection of a few new icebergs for each lower threshold level, while in the second phase, the rapid increase
380    is likely triggered by speckle or noise within our search radius. We will discuss this issue more in detail in Sect. 5.2.

Comparing the recall for the two sensors, we found that all six detectors behave very similarly when applied to the L- and C-band images for low PFA-levels, with the exception being the Log-normal detector which show a higher recall for the L-band image. At higher PFA-levels the recall for Sentinel-1 increases earlier and more strongly compared to ALOS-2. For a PFA





level of 0.1, all detectors in the Sentinel-1 image are detecting 97%-99% of the icebergs, compared to only 79%-97% for ALOS-2 (but note that the probability of false detections is very high, see Fig. 5). The NIS, Wishart, and Ganna detectors give the highest recall for both L- and C-band, whereas K- and Log-normal detectors generally show a poor recall for low to medium PFA-levels.

Our results for the precision shown in Figure 5 reveal that most of the detectors have a constant or slightly decreasing precision for small PFA-levels. This trend is caused by a small increase in the number of false positives for each lower threshold level. With increasing PFA-levels, the precision reaches a point where it starts to rapidly decrease towards zero, corresponding with a large increase in the number of false positives coupled with only a small increase in true positives. When the threshold becomes very low, intensity variations due to speckle will start triggering the detector, causing an increased number of false positives, which leads to a rapid decrease of precision. When comparing the two sensors we found a very similar performance for the Gamma, Log-normal and K detectors for C- and L-band. Where especially the Log-normal and K detectors show a very high precision over a wide range of smaller PFA-values. This high precision at low PFA-levels is mainly driven by the fact that these detectors only detect a small number of large and bright icebergs, with almost zero false positives. However, this also means that these detectors have a high number of false negatives, which is also evident when comparing the precision (Fig. 4) with the recall (Fig. 5). The Wishart, NIS and iDPolRAD all show lower precision at L-band compared to C-band, suggesting noise in the L-band image is triggering these detectors. Especially the iDPolRAD detector show a very large decrease in precision. This indicates that in our data set, more spots of strong backscattering in HV not caused by icebergs occur at L-band than at C-band.

In general, high recall comes with low precision. This makes sense, as there is an overlap between the intensity backscatter distributions for open water and icebergs, and a detector that is capturing more icebergs, will therefore likewise capture more spots of strong backscattering from the water surface as well. The exception here is the iDPolRAD filter, which for ALOS-2 shows very low precision which suggests that this detector is being triggered more often by noise than the other detectors.

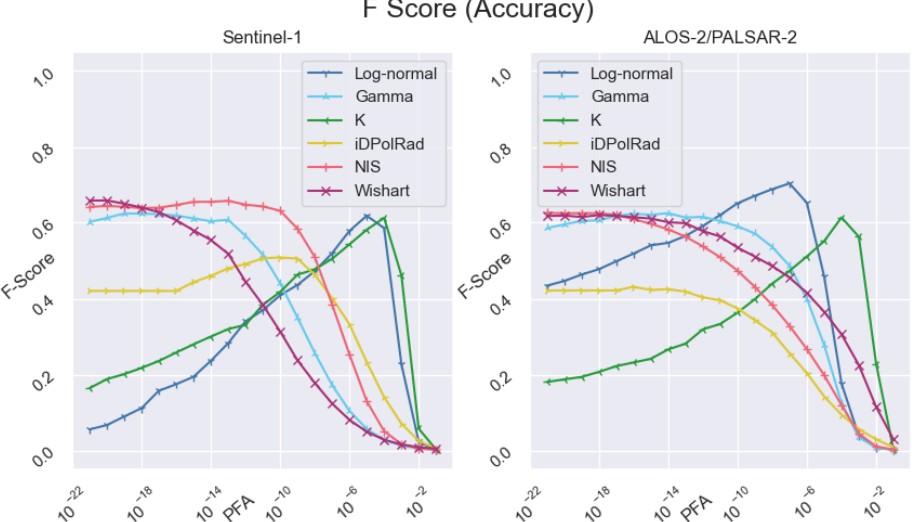

**Figure 6 F-score – The overall accuracy of the different detectors as a function of the PFA**

The F-score is shown in Fig. 6 combines recall and precision which makes an overall assessment of the different detectors much easier. In general, most of the detectors show poor F-scores for very small PFA-levels due to low recall, and for high PFA-levels due to low precision. Between those extremes the F-score is at maximum, corresponding to a region where we obtain the optimum balance between missed and false detections. However, the exact point of this optimum PFA level is very





different for the different detectors and varies also between C- and L-band The large differences in the optimal PFA level for a given detector highlights the importance of determining recall and precision at different PFA levels.

Overall, the performance of iceberg detection accuracy of Sentinel-1 and ALOS-2 is comparable, with each sensor obtaining maximum F-scores of around 60%-70% for all detectors except for the iDPolRAD which generally scores lower. This issue is discussed in Sect. 5.

The K-detector and the Log-normal detector achieve their maximum accuracy at a very narrow interval of PFA-levels, whereas the gamma, iDPolRAD, NIS, and Wishart generally exhibit higher F-score across a wide range of PFA-levels. The comparisons between the shapes of the F-score as a function of PFA reveals that there is no single detector achieving best performance over the whole range of PFAs. Results obtained at C- and L-band show that one detector may be optimal at one frequency but another detector for the other frequency. For some filters it is better to select a lower value of PFA, for others a higher value leads to higher F-scores. Nevertheless, Figs. 4, 5, and 6 are helpful in deciding which detector to use for a given PFA and radar band. Below, we include additional criteria that should be considered in the selection of a specific filter. It must be admitted, however, that the accuracy of detection is still too low for an unsupervised mapping of iceberg positions to be used for navigation. Further experiments with different modes of SAR images and combinations of images acquired simultaneously at different radar frequencies are required.

**Table 3 Sentinel-1. Number of false negatives (FN), false positives (FP), and true positives (TP) for each of the detectors at the PFA-level resulting in the highest F-score.**

|        | Gamma (1e-19) | iDPolRAD (1e-10) | K (1e-4) | Log-normal (1e-5) | NIS (1e-13) | Wishart (1e-20) |
|--------|-------|----------|-----|------------|------|----------|
| **FN** | 254   | 299      | 253 | 261        | 219  | **216**  |
| **FP** | 32    | 74       | 49  | **24**     | 65   | 70       |
| **TP** | 238   | 193      | 239 | 231        | 273  | **276**  |

**Table 4 ALOS-2. Number of false negatives (FN), false positives (FP), and true positives (TP) for each of the detectors at the PFA-level resulting in the highest F-score.**

|        | Gamma (1e-14) | iDPolRAD (1e-16) | K (1e-4) | Log-normal (1e-7) | NIS (1e-21) | Wishart (1e-18) |
|--------|-------|----------|-----|------------|------|----------|
| **FN** | 230   | 285      | 255 | **201**    | 217  | 215      |
| **FP** | 84    | 261      | **43** | 44      | 111  | 124      |
| **TP** | 262   | 207      | 237 | **291**    | 275  | 277      |

For each of the detectors, results from the optimal configuration, i.e., the PFA-level that resulted in the highest F-score, were extracted. The total numbers of false negatives, false positives, and true positives are shown in Table 3 for Sentinel-1 and in Table 4 for ALOS-2. For Sentinel-1, the Wishart detector obtained the highest number of true positives and the smallest number of false negatives, and for ALOS-2 this was obtained by the Log-normal detector. For C-band, the smallest number of false positives was achieved by the Log-normal detector, and for L-band by the K detector, but with the Log-normal detector showing only a single additional false positive. It is worth noting that the number of false negatives is almost as large as of the true positives for all detectors, which means that almost half of the icebergs are missed by all the detection algorithms. To investigate why so many icebergs are missed, we have added the detection accuracy for various iceberg sizes below.





**Table 5 Sentinel-1. Percentage of detected icebergs for each of the detectors at the PFA-level resulting in the highest F-score.**

| Icebergs | N | Gamma (1e-9) | iDPolRAD (1e-10) | K (1e-2) | Log-normal (1e-2) | NIS (1e-13) | Wishart (1e-20) |
|----------|-----|-------|-------|-------|-------|-------|--------|
| **Small** | **181** | 6.6% | 5.0% | 6.6% | 6.6% | 11.6% | **13.3%** |
| **Medium** | **175** | 57.7% | 40.6% | 58.9% | 56.0% | **70.3%** | 69.7% |
| **Large** | **136** | 91.9% | 83.1% | 91.2% | 89.0% | 94.9% | **95.6%** |

**Table 6 ALOS-2. Percentage of detected icebergs for each of the detectors at the PFA-level resulting in the highest F-score.**

| Icebergs | N | Gamma (1e-7) | iDPolRAD (1e-16) | K (1e-2) | Log-normal (1e-4) | NIS (1e-21) | Wishart (1e-18) |
|----------|-----|-------|-------|-------|-------|-------|--------|
| **Small** | **181** | 10.5% | 5.5% | 7.7% | **15.5%** | 9.9% | 11.0% |
| **Medium** | **175** | 65.1% | 48.6% | 57.1% | **80.6%** | 73.1% | 72.0% |
| **Large** | **136** | 94.9% | 82.4% | 90.4% | 89.7% | 94.9% | **96.3%** |

Of the 492 icebergs used in the study, 181 are classified as small, 175 as medium, and 136 as large (Table 2). For each of the detectors, the optimum PFA value was chosen to determine the absolute number of detected icebergs as a function of iceberg size for Sentinel-1 (Table 5) and ALOS-2 (Table 6). The results clearly demonstrate the considerable increase of detection rates for larger icebergs and hence the influence of the effective spatial resolution of the SAR images. For small icebergs, detection rates are extremely low, emphasizing the need for employing SAR systems which acquire images with high spatial resolutions on the order of 10 m (while at the same time keeping a large swath width). As icebergs become smaller than the resolution limit of the sensors, most icebergs are no longer separable from the background clutter. A few small icebergs are still detected, which might be due to their orientation or geometry giving rise to a strong backscattering into the direction of the SAR antenna.

**Table 7 Execution times of the different detection algorithms. The speed test was done for a 1000x1000 pixel subset of the Sentinel-1 EW scene (left), and for the full Sentinel-1 EW image (right) of approximately 110x10^6 pixels *For the K detector, calculation of the look-up table takes approximately 20 seconds**

| Algorithm | Run time (1000x1000 px) | Run time (entire S1EW scene) |
|-----------|-------------------------|------------------------------|
| Log-normal | 292 [ms] | 41.3 [s] |
| Gamma | 144 [ms] | 20.1 [s] |
| K-Distribution* | 2157 [ms] | 76.0 [s] |
| iDPolRAD | 1319 [ms] | 181.4 [s] |
| NIS | 185 [ms] | 27.6 [s] |
| Wishart | 174 [ms] | 27.8 [s] |


The execution time for object detection is an important issue in operational iceberg mapping. Therefore, we determined it for the six detectors. Each detector was run at a PFA-level of 1e-12, and the execution time was tested both on a small subset of 1000x1000 pixels, and on the whole Sentinel-1 EW scene. The small subset covered an area containing only water and icebergs, but for the whole Sentinel-1 EW scene, about 10% of the image was masked as being land before applying the



detectors – leaving approximately 110x10e^6 pixels to be analyzed. The test was carried out on a 64-bit PC, equipped with
an i7 processor @ 2.60GHz and 32.0 GB RAM. Four detectors could be run on a full Sentinel-1 EW scene in less than a
minute, making them well suited for operational applications. For the K detector, the execution time was 76 seconds, but
around 20 seconds is attributed to the look-up table calculation which can be carried out once and afterwards be re-used in
operational systems.




(a) Log-normal Detector at PFA of 1e-2

(b) Gamma Detector at PFA of 1e-9

(c) K Detector at PFA of 1e-2

(d) iDPolRAD Detector at PFA of 1e-10

(e) NIS Detector at PFA of 1e-13

(f) Wishart Detector at PFA of 1e-20

**Figure 7 Detections for the 6 detectors for a subset of the C-band Sentinel-1 image. Green circles: True Positives. Yellow squares: False Negatives. Red triangles: False Positives. HV in red channel, HH in green and blue channel. The background SAR composite is color coded as Red: HV, Green: HH, Blue: HH.**







**(a) Log-normal Detector at PFA of 1e-4**

**(b) Gamma Detector at PFA of 1e-7**

**(c) K Detector at PFA of 1e-2**

**(d) iDPolRAD Detector at PFA of 1e-16**

**(e) NIS Detector at PFA of 1e-21**

**(f) Wishart Detector at PFA of 1e-18**

**Figure 8 Detections for the 6 detectors for a subset the L-band ALOS-2 image. Green circles: True Positives. Yellow squares: False Negatives. Red triangles: False Positives. HV in red channel, HH in green and blue channel. The background SAR composite is color coded as Red: HV, Green: HH, Blue: HH.**




The performance of the different detectors was further assessed by visualizing the results on subsets of the SAR images which are shown in Fig. 7 and Fig. 8 for Sentinel-1 and ALOS-2, respectively.

The subset covers an area containing several icebergs grounded along the coast of Labrador, as well as several icebergs floating
in the open water. Objects identified with the various detectors are marked on the figures, with green triangles denoting True Positives (TP), yellow squares False Negatives (FN), and red circles False Positives (FP). The same subset is shown for Sentinel-1 and ALOS-2 images, but due to the different acquisition times, the position of some icebergs has changed between the two images. Both images were acquired in ScanSAR mode, and the subsets cover the border between two subswaths, which gives rise to a diagonal line with a different signal-to-noise ratio (SNR) on each side. This is seen in the top left corner of the
Sentinel-1 image, and through the center of the ALOS-2 image. In the Sentinel-1 image, variations of the sea clutter are visible as brightness differences in the open water. In the ALOS-2 image bright artifacts occur in the water, likely caused by range- and azimuth-ambiguities from the processing. The high density of icebergs, various clutter states, and image artifacts, makes this subset well suited to illustrate the advantages, and disadvantages, of the various detectors.

Figure 7 reveals that the Gamma, Log-normal, and K-detectors all behave similarly, with a limited number of false positives,
and an approximately equal number of true positives and false negatives. The NIS and Wishart detectors both show a lower number of false negatives along the coast in the top left of the image, but also show a higher number of false positives in the top center of the image. These false detections appear to be caused by linear features in the open water, possibly some sort of ocean waves which triggers false detections. With the iDPolRAD we obtained many false positives along the boundary between the subswaths. This area is also characterized by increased levels of noise in the HV-band (red color), which may
falsely trigger the detector. Additionally, the iDPolRAD detector also shows many missed detections compared with the other detectors, especially along the coast in the top left corner of the subset.

For the ALOS-2 image in Figure 8, we found a very similar performance for the Log-normal and K detectors. Both show a low number of false positives, whereas more false positives are obtained with the Gamma, NIS, and Wishart detectors along the diagonal subswath boundary through the center of the image. These three detectors also identified an increased number of
false positives in the bottom of the image, which appear to be caused by azimuth ambiguities. As in the Sentinel-1 image, the iDPolRAD shows the highest number of false positives.

### 4.1 Summary

The highest f-score (70.4%) was achieved by the Log-normal detector on the ALOS-2 image. Besides this detector, the performance of the different methods for iceberg detection is similar on C- and L-band SAR images, with maximum f-scores
around 60-65%. However, we note that we had data available only for a case study of icebergs in open water under low wind speeds and melting conditions. Hence, further comparisons between L- and C-band may show that SAR images acquired at a particular frequency should be used for conditions not investigated here.

When comparing the results for the F-score as a function of PFA obtained for the investigated detectors, we found that all are suitable for use considering individual limits, with the exception of the iDPolRAD detector which can be related to the melting
conditions (see Sect. 5). The advantage of the Gamma, NIS, and Wishart detectors provide high F-scores over a larger range of small PFA values. In wider range of low PFA-levels we found an almost constant F-score also for the iDPolRAD. The Log-normal and K-detectors showed a narrow maximum of the F-score at larger levels of the PFA, which means that the thresholds for the optimal PFA are more difficult to fix beforehand. Moreover, according to our results, they differ between C- and L-band. We expect that the position of the maximum F-score on the PFA axis may also depend on the specific conditions
(freezing, melting, rough or smooth water and ice surface).

Our results of the detectability as a function of iceberg size shows that as many as 20-40% of the medium icebergs (60-120 meters) are not found using the detection methods tested here, even though the pixel spacing of the images used is of 40 meters. This indicates that many medium-sized icebergs might be missed in operational charting.



**5 Discussion**

**5.1 Factors influencing the accuracy of detection rates**

For verifying the iceberg detections in the SAR images based on matches with icebergs identified in the Sentinel-2 image, we had to consider the drift of the icebergs between the times of acquisitions of the different images as explained above. Larger icebergs could be identified more easily in all images. For those icebergs, we used the direct displacement between the respective SAR image and the Sentinel-2 image as drift vector. For icebergs shorter than 50 meters which were more difficult

to match between the optical and the radar images, we estimated a drift path based on an interpolation between adjacent drift vectors from larger icebergs. The interpolation builds on the assumption that the smaller icebergs maintain the same heading and speed as the neighbouring larger icebergs. This assumption might not always hold considering that the forces from wind and currents on icebergs depend on the cross-sections of their sails and keels (Wesche and Dierking, 2016), which causes larger uncertainties of the drift vectors, especially over large distances. Considering that the drift field between the Sentinel-1

and the Sentinel-2 image contains more interpolated drift paths and that the drift distance is larger, we may underestimate the performance of iceberg detection at C-band. In the future, more advanced drift predictions could be used to limit this type of uncertainty. Alternatively, having optical data acquired at the same time as the SAR overpasses would avoid the need of advanced drift correction. In our case, we also must take into account that the ALOS-2 images were down-sampled, which means that the identification of icebergs is made more difficult in the L-band images.


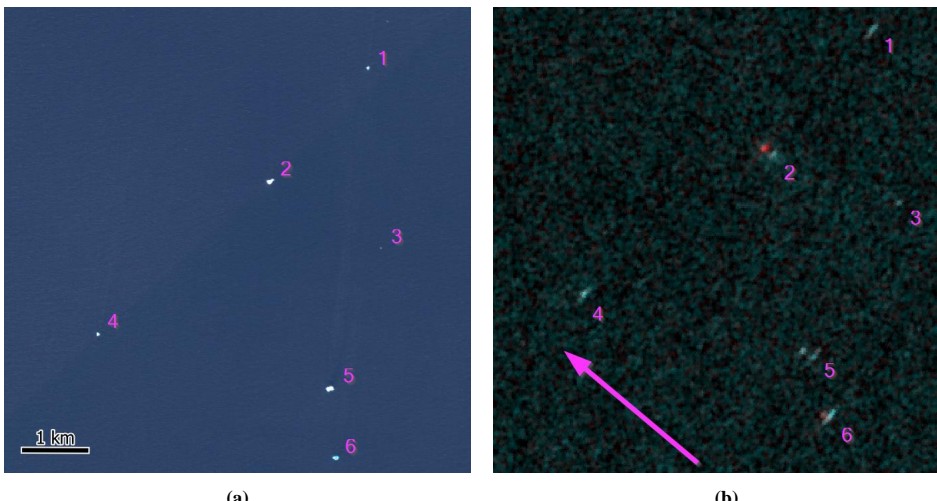

(a)                        (b)

**Figure 9 Example of the internal ghost reflections. Six icebergs are marked in the Sentinel-2 image (a), and in the ALOS-2 image (b). Icebergs number 2, 5, and 6 show two distinct reflections, with an initial bright reflection followed by a reflection further down-range. The secondary reflection is typically dimmer and often dominated by HV scattering (red). The look direction of the ALOS-2 satellite is marked by the arrow.**

Other factors that influence the results which we obtained for missed and false detections are iceberg disintegration (icebergs breaking into smaller fragments) and the occurrence of. ghost reflections at L-band (Gray and Arsenault, 1991). As illustrated in Fig. 9, ghost reflections were visible in the ALOS-2 imagery as small secondary reflections down-range of the main radar returns for some icebergs.

The effect of different wind conditions and sea states on the detection performance could not be systematically investigated in

our study since the needed simultaneously acquired images at C- and L-band are still lacking. It is expected that L-band SAR is less sensitive to sea surface roughness than C-band, and hence L-band should in theory be better suited for iceberg detection under rough wind conditions (Dierking, 2020). Increasing wind also may increase the variation of backscattering (affecting





the image texture), which could have the positive effect of increasing the accuracy for the K detector. Similarly, increasing wind conditions might lead to worse performance for the detectors based on the Gamma distribution, i.e., the Gamma, NIS, and Wishart detectors. For future work, it is of great interest to investigate how various detectors perform under varying wind conditions, and to consider different radar frequencies and polarizations. In our analysis, we also found a high number of false detections due to the noise occurring at the boundaries between subswaths, as well as due to range and azimuth ambiguities near the coast. In an operational system, range and azimuth ambiguities can be filtered during pre-processing, or masked from the study area. Similarly, detections along the border between two subswaths can also be removed in post-processing.

### 5.2 Interpretation of results for the recall

As mentioned earlier, the rapid increase in recall with increasing PFA may be caused by the way we match icebergs between the optical and the radar image. To account for uncertainties, we define a search radius in the radar image around the expected position of an iceberg identified in the optical image, based on its estimated drift path. Especially for the interpolated drift vectors, this search radius is large (Sect. 3.4). So as the PFA-level increases, the detectors will more and more be triggered by speckle or other radar intensity variations (e.g., due to strong reflections from a rough water surface) within the search radius. Since the search radius is larger for Sentinel-1, and Sentinel-1 also has a lower ENL, and hence higher variance of the speckle, we expect that this issue will be more obvious for Sentinel-1 compared to ALOS-2, which may explain the sudden large increase in recall for Sentinel-1. As the number of false detections increases inside the search circle, it will also increase outside, which in turn will lead to a sharp decrease in precision. Both effects determine the shape of the F-score curves shown in Fig. 6.

### 5.3. Interpretating the results of different detectors

The different detectors show a maximum of the F-score at different PFA-levels: the NIS, Gamma, and Wishart detectors at very low PFA-levels, and the K- and Log-normal detectors at higher PFA-levels. Looking at the recall (Fig. 4) and precision (Fig. 5), we see that the K- and Log-normal detectors are generally less sensitive to both icebergs and noise (late increase in recall and late decrease in precision). We found that the different detectors performed similarly for the two SAR sensors, except for the Log-normal detector, which showed a higher F-score for L-band than for C-band. This could indicate that the model of radar intensity variations due to sea clutter used in the Log-normal detector is more accurate at L-band than at C-band at least for the sea states that were represented in our dataset.

The similar overall performance of the Gamma, NIS and Wishart detectors is a consequence of the design of these detectors. All three assumes that the noise in the HH and HV channels follows a gamma distribution. The NIS detector finds outliers in the linear combination of the normalized intensities at HH and HV polarization. The Wishart detector is based on the multi-dimensional Wishart distribution in which the contributions of the different polarizations are considered. And the Gamma detector find outliers in the HH and HV bands independently and highlights outliers visible in both bands. Hence, all three detectors should be sensitive to the same outliers, with some minor variations for targets that show an increased backscatter for one of the polarizations only.

In general, the iDPolRAD detector shows only moderate performance across all performance measures and for both C- and L-band. The main weakness of the detector appears to be its tendency to be triggered by the noise occurring along the subswath boundaries. However, these false detections could theoretically be filtered in the post-processing. The reason we did not apply a filter like this for our study, was possible complications a filter like this might add, e.g., a subswath filter might remove some true positives.

Another weakness of the iDPolRAD detector is its sensitivity to melting conditions. Melting at the surface of icebergs causes a considerable decrease of the penetration depth of the SAR signal at both L- and C-band, hence reducing the volume scattering component. The presence of the latter is the major criterion for separating icebergs from the background dominated by surface



scattering. Additionally, since the icebergs used in this study have drifted far from their calving areas, they may have turned over underway, causing the formation of salty ice layers on the icebergs' surface, which also leads to a decrease of volume scattering. The detector should hence only be applied under freezing conditions, and close to calving sites, where rolling-over is less probable.

### 5.4. Sensitivity of F-score to changes of PFA

The main strength of using the F-score is that it reveals how changes in PFA influence the accuracy of detection. We clearly
demonstrated that some detectors are quite sensitive to small changes in PFA-level, whereas others are quite stable under varying PFA-levels (Fig. 6). Normally, detectors are compared at similar PFA-levels because it is assumed that the PFA is representative of the actual false alarm rate, but due to various implementation details or inaccurate assumptions as mentioned earlier, this might not always be the case. Our results showed that the NIS, Gamma, and Wishart detectors obtaining a high accuracy across a wide range of PFA-levels. Similarly, the Log-normal detector obtained a high f-score for a wide range of
PFA-levels for the ALOS-2 data. But K detector (and the Log-normal detector applied to C-band) obtained a high accuracy only for a narrow range of PFA values. This could indicate that these distributions are ill-suited for fitting variations of the sea surface clutter in the tail of the distribution, so that small changes in PFA-level result in large changes in the cut-off threshold for determining whether pixels belong to the background or iceberg class. However, more work is required to confirm this. Nevertheless, our results underline the importance of looking at a broad range of PFA-levels instead of evaluating all the
detection algorithms at one fixed PFA-level.

### 6 Conclusion

In this paper, we have compared the performance of 6 different CFAR detectors for iceberg detection in both a C-band Sentinel-1 SAR image, and a L-band ALOS-2 SAR image. Both images were acquired over the same region of the Labrador Sea and were acquired in wide-swath dual-polarization (HH/HV) mode. A total of 492 icebergs were visually identified in the study
area, using an overlapping Sentinel-2 image for validation. The performance of the detectors was assessed by counting the number of false positives, true positives, and false negatives, and calculating the corresponding recall, precision, and F-score. Each detector was tested at varying PFA-levels, making it possible to assess the performance of the detectors as a function of the PFA level. Additionally, the results for the PFA-level that gave the highest F-score were analyzed to investigate the detection rate for icebergs at varying sizes.
Comparing the individual detection algorithms on C- and L-band revealed no large differences, except for the iDPolRAD detector which showed a higher f-score on C-band, and the Log-normal detector which showed a higher f-score on L-band. This shows that not all detectors tested on C-band imagery, can be applied to L-band imagery with the same expected results. Overall, the highest accuracy was obtained by applying a Log-normal CFAR detector to the ALOS-2 L-band image, which gave an F-score of 70.4%. In general, the Gamma, NIS, and Wishart detectors all gave F-scores above 62% for both C- and L-
band. Additionally, these three detectors showed to be very stable to changes in the PFA-level. The K detector resulted in F-scores comparable with the other detectors but also showed to be very sensitive to tuning of the PFA-level. A similar result was obtained by the Log-normal detector applied to the C-band image. A detector developed for iceberg detection in sea ice, the iDPolRAD detector, showed only moderate performance for icebergs in open water – possibly due to the high temperatures in the study area.
Three different methods were tested for merging the dual-channel HH/HV images for CFAR detection. But the methods used did not appear to give rise to any significant differences. With similar performance for the Gamma, NIS, and Wishart detectors. Only 10-15% of the icebergs shorter than 60 meters could be detected in the dataset, suggesting that wideswath SAR images at both C- and L-band are insufficient for detecting small icebergs. Additionally, between 20-50% of the medium icebergs (60-



120 meters), and 5-20% of the large icebergs (>120 meters) were missed by the detectors. This shows that a large part of icebergs that are 1.5-3 times size of a single pixel are not being detected, suggesting a risk of underestimating iceberg conditions by operational iceberg charting services.

Each of the detectors obtained their highest F-score at different PFA-levels. This suggests that comparing detectors at the same PFA-level will give inaccurate results. The results also revealed that some detectors were sensitive to variations in PFA-level while others proved more stable. This suggests that the sensitive detectors should be used with care or undergo manual tuning for optimum results. We therefore recommend that the detectors with stable response to changing PFA–level, namely NIS, Wishart and Gamma, are used when implementing an operational iceberg detection product.

L-band appear to offer a slight improvement over C-band on the dataset in this study, we expect this improvement to be even higher for cases with more wind, and we encourage further investigations of the use of L-band SAR data for detecting icebergs under varying wind conditions.

*Code Availability.* Implementations of the different detectors are available on GitHub ([https://github.com/LaustFaerch/cfar-object-detection](https://github.com/LaustFaerch/cfar-object-detection)).

*Author Contribution.* The method was developed by LF and WD. The dataset was produced and analyzed by LF, who also led the manuscript writing. APD gave advice on the methodology and on the implementation and theory behind the detectors. NH offered advice on the operational aspects of iceberg detection services. WD, APD, and NH all contributed to the manuscript.

*Competing interests.* The authors declare that they have no conflict of interest.

*Acknowledgements:* ALOS-2/PALSAR-2 data are provided by JAXA through the 2019 to 2022 mutual cooperation project between ESA and JAXA on Using Synthetic Aperture Radar Satellites in Earth Science and Applications.

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
