# Peer review of "A Comparison of CFAR Object Detection Algorithms for Iceberg Identification in L- and C-band SAR Imagery of the Labrador Sea"

_The Cryosphere, 2023_

## Author Response (AR1)

**Comments from Anonumous Referee #1**

The preprint version of the paper "A Comparison of CFAR Object Detection Algorithms for Iceberg Identification in L- and C-band SAR Imagery of the Labrador Sea" is of very high quality and ready for publication. The study is explained in a comprehensive and clear way, figures and tables are all fine.

The only weakness in the study presented in the preprint is that all tests were performed on only one C and one co-located L-band SAR image, i.e. they capture the same sea state and wind conditions. That is, the ocean backscatter does not vary much. Since the core of CFAR algorithms is based on the evaluation of ocean backscatter (and certain backscatter distributions such as e.g. gamma, log-normal), using only one image for each band is not the best basis for comparing different CFAR algorithms.

*We agree that only using a single image pair for comparison is a weakness of the paper. The reason for only using a single image pair is 1) the limited number of ALOS-2 scenes we had available (very few of the ALOS-2 images we had showed a large number of icebergs while also simultaneously being acquired in a day where we had a Sentinel-1 image and a mostly cloud-free optical image available). 2) the amount of manual work required to do the matching of the drift vectors between the different images.*

*Changes:*

1. *We have added a paragraph at line 547-550 mentioning that since only a single image pair was used, the influence of different wind and sea states could not be investigated. However, due to the large number of icebergs used in the study, the results are still valid under low wind conditions.*

**Comments from Anonymous Referee #2**

Summary

This paper does a nice good of using and summarizing the results of using 6 different CFAR algorithms for detecting icebergs with both L- and C-band data. The validation data is a Sentinel-2 image. All 3 data sets are obtained on the same day within a few hours of each other under low to moderate wind speeds (1-5 m/s) and temperatures above freezing. The differences in algorithms, including strengths and weaknesses of each one, are nicely described. I recommend minor revisions to be considered for inclusion in the final paper.

Detailed comments

1. As pointed out in the previous review, yes its unfortunate that only two SAR images are utilized. However, this is balanced by the very useful Sentinel-2 image with the large number of icebergs identified and the fact that all 3 data sets were obtained on within hours on the same day. It is clear that lots of work went into assessing these images.

*We have added a comment in the discussion addressing the issues with using only a single image pair, with limited environmental conditions, but our claims with respect to the different frequencies are unaffected.*

2. I recommend considering using or evaluating some additional pre-processing factors. Could better speckle filtering be used to reduce speckle and then false detection? Also it is likely that the backscatter differences at image seams between ScanSAR subswaths could be further minimized in pre-processing.

*Speckle filters tend to reduce the spatial resolution and in some cases are known to add artefacts. This makes them ill-suited for iceberg detection. When applying speckle filtering, one would have to consider the balance between e.g., spatial resolution, detectable iceberg size, finding an optimal speckle filtering method, and the degree of filtering. A systematic investigation of this is beyond the scope of this paper but could be the topic for another study where the effect of speckle filtering on the detection rate is investigated for a single CFAR detector.*

*We are aware of some methods that could reduce the noise and intensity jumps between subswaths. E.g., [1] and [2]. But these methods remain to be extensively tested and require careful analysis, e.g., in terms of their effect to small point targets. An analysis like this is also beyond the scope of this study.*

*We have added a comment about how noise reduction methods might reduce the number of false positives along subswath transitions, but the methods need to be investigated more in depth before being applied for iceberg detection.*

*[1] W. Yang, Y. Li, W. Liu, J. Chen, C. Li and Z. Men, "Scalloping Suppression for ScanSAR Images Based on Modified Kalman Filter With Preprocessing," in IEEE Transactions on Geoscience and Remote Sensing, vol. 59, no. 9, pp. 7535-7546, Sept. 2021, doi: 10.1109/TGRS.2020.3034098.*

*[2] J. -W. Park, J. -S. Won, A. A. Korosov, M. Babiker and N. Miranda, "Textural Noise Correction for Sentinel-1 TOPSAR Cross-Polarization Channel Images," in IEEE Transactions on Geoscience and Remote Sensing, vol. 57, no. 6, pp. 4040-4049, June 2019, doi: 10.1109/TGRS.2018.2889381.*

3. The consideration of the noise floors of each sensor at both polarizations should also be evaluated before use since the wind speeds seemed to be fairly low and the noise floors vary by frequency and polarizations. I recommend looking at other remote sensing data that would indicate surface wind speed near the time of acquisition of each data set, including Sentinel-1 derived wind speeds distributed by both NOAA and ESA-related groups. The ALOS-2 frame was only 1 hour later than the S-1image so that the winds are likely be similar even if only using S1 winds for example. This more precise data should be useful for addressing items discussed above regarding noise floor and signal to noise ratio. The HH/HV ratios may be quite different at L and C especially with incidence angle. Consider the use of the paper by Espeseth et al (2020) in TGARSS.

*We have added information about the wind speed from the Sentinel-1 OCN product. There are 6 hours between Sentinel-1 and ALOS-2 acquisitions, which means that the wind could have changed in the meantime. But it appears as if there was low wind in the area during the entire day. We confirmed this using data from ECMWF.*

*In case of very low wind, the ocean backscatter will be influenced by the noise floor. For Sentinel-1, the noise floor as a function of incidence angle is well known. We have not been able to find a detailed documentation for the ALOS-2 noise floor, except the mission requirements which states that the noise equivalent sigma zero should be always better than -26 dB. For areas where the ocean backscatter is close to or below the noise floor, there is a chance of false positives caused by the noise. This is e.g., seen in our results as a large number of false positives along subswath transitions both for ALOS-2 and Sentinel-1.*

*We have added a comment outlining how the number of false positives found along the subswath boundaries might be higher than expected due to low wind and noise floor variability.*

4. As mentioned in the conclusions, imagery from higher winds should be very useful, for the reasons provided. It may also be useful to consider using imagery obtained during colder temperatures and further north. Such data would likely contain more sea ice which may be problematic for positive detection. Using two or more pairs of images would enable drift estimates and perhaps these could be used as Lagrangian drifters in ocean current models as the icebergs move into warmer conditions. As the authors point out, sea ice will melt likely melt sooner than ice bergs.

*The issue with looking at icebergs in open ocean surrounded by sea ice, is that it is very hard to independently validate from optical imagery which objects are icebergs and which are sea ice. In addition, positively identifying individual icebergs in a time-series of images is difficult, especially considering the large time-difference needed to monitor icebergs as they drift from the colder areas with sea ice down to warmer areas of open water, and also considering that the icebergs are changing due to melting, and pieces breaking off.*

*We agree that it will be useful to consider imagery obtained during colder temperatures and in the presence of sea ice. We have a paper in review addressing some of these challenges by looking at icebergs embedded in land-fast sea ice during both freezing and melting temperatures.*

*We have added a comment in the discussion, mentioning that the results might be different under freezing conditions.*

5. The detection of smaller icebergs is definitely the key issue to try and improve upon, especially as the bergs move into open ocean and drift into shipping lanes. Keep after the problem.

*We have added a comment in the discussion pointing at possible solutions for improving the detection accuracies of small icebergs: 1) developing better models for clutter estimation. 2) investigating the use of single-look images and the trade-off between noise and resolution. 3) using SAR sensors with higher spatial resolution.*

6. Lines 58-60. I suggest including a mention of NISAR since it is planned to launch in early 2024 well before ROSE-L and will include coverage of all sea ice cover including off Labrador. https://nisar.jpl.nasa.gov/mission/observation-strategy/

*We have added a comment regarding NISAR in the introduction.*

7. I suggest including both SAR images as figures in the paper, at least to give a perspective on the ocean backscatter.

*We have updated Figure 1 so that it includes both SAR images.*

*Changes:*

1. *We have added a paragraph at line 547-550 mentioning that since only a single image pair was used, the influence of different wind and sea states could not be investigated. However, due to the large number of icebergs used in the study, the results are still valid under low wind conditions.*

2. *We have added a paragraph from line 558-562 mentioning possible noise reduction methods that could be considered in operational systems.*
3. *We have added a paragraph from line 551-558 addressing the relationship between wind, system noise and false detections.*
4. *We have added a comment at line 569-570 mentioning that colder temperatures might affect the results.*
5. *We have added a paragraph at line 571-578 mentioning possible solutions for a more accurate detection of small icebergs.*
6. *We have added a comment at line 59-63 mentioning NISAR mission.*
7. *We have updated Figure 1 so it includes both SAR images.*

*Additional Changes:*

1. *Sub-captions removed from Figure 3 so that the figure only have one caption now.*
2. *Tables 3-6 Ambiguous notation for exponents changed.*
3. *Table 7 Ambiguous notation for exponents changed in caption.*
4. *Section 4. Ambiguous notation for exponents changed in text.*
5. *Figures 7-8 Ambiguous notation for exponents changed in captions.*
6. *Acknowledgement to our partners at ESA and MOST added.*
7. *References added.*